**Validation of SOFIE nitric oxide measurements**
**Mark E. Hervig[1],*, Benjamin T. Marshall[2], Scott M. Bailey[3], David E. Siskind[4], James M.**
**Russell III[5], Charles G. Bardeen[6], Kaley A. Walker[7], Bernd Funke[8]**
[1]GATS, Driggs, Idaho, USA.
[2]GATS, Hampton, Virginia, USA.
[3]Virginia Polytechnic Institute, Blacksburg, Virginia, USA.
[4]Space Science Division, Naval Research Laboratory, Washington, DC, USA.
[5]Hampton University, Hampton, Virginia, USA.
[6]NCAR, Boulder, Colorado, USA.
[7]Department of Physics, University of Toronto, Toronto, Canada.
[8]Instituto de Astrofísica de Andalucía, CSIC, Granada, Spain.
*Corresponding author. E-mail address: m.e.hervig@gats-inc.com (M. Hervig).
**Abstract.**    Nitric oxide (NO) measurements from the Solar Occultation for Ice Experiment
(SOFIE) are validated through detailed uncertainty analysis and comparisons with independent
observations. SOFIE was compared with coincident satellite measurements from the Atmospheric
Chemistry Experiment (ACE) - Fourier Transform Spectrometer (FTS) instrument, and the
Michelson Interferometer for Passive Atmospheric Sounding (MIPAS) instrument. The
comparisons indicate mean differences of less than ~50% for altitudes from roughly 50 to 105 km
for SOFIE spacecraft sunrise, and 50 to 140 km for SOFIE sunsets. Comparisons of NO time series
show a high degree of correlation between SOFIE and both ACE and MIPAS for altitudes below
~130 km, indicating that measured NO variability in time is robust. SOFIE uncertainties increase
below ~80 km due to interfering $H_2O$ absorption, and from signal correction uncertainties which
are larger for spacecraft sunrise compared to sunset. These errors are sufficiently large in sunrises
that reliable NO measurements are infrequent below ~80 km.

## 1. Introduction

The Solar Occultation for Ice Experiment (SOFIE) has measured nitric oxide (NO) from
the Aeronomy of Ice in the mesosphere (AIM) satellite since May 2007. SOFIE NO measurements
have been the topic of numerous science investigations, including studies of thermosphere -
stratosphere coupling (Bailey et al., 2015; Siskind et al., 2015; Hendrickx et al., 2018), effects of
the 27-day solar rotation (Hendrickx et al., 2015), and the roles of dynamics and chemistry in
diurnal variability (Siskind et al., 2019). SOFIE NO observations have also been used to determine
the importance of changes in geomagnetic activity and solar radiation (Hendrickx et al., 2017),
and to characterize the response of NO to electron precipitation (Smith-Johnsen et al., 2017; 2018;
Newnham et al., 2018). SOFIE version 1.3 (V1.3) NO measurements are validated here through
uncertainty analysis and comparisons with correlative measurements.
Coincident satellite measurements are from the Atmospheric Chemistry Experiment (ACE)
- Fourier Transform Spectrometer (FTS) instrument, and the Michelson Interferometer for Passive
Atmospheric Sounding (MIPAS) instrument. The ACE-FTS instrument has used solar occultation
to measure more than 30 trace gases and over 20 isotopologues from 2004 to present (Bernath et
al., 2005). ACE NO measurements span ~6 to 107 km altitude with a vertical resolution of ~3.5
km, and retrievals are reported at the oversampled vertical interval of 1 km. This work used version
3.5 NO retrievals, which are based on measurements between 5.056 and 6.063 μm wavelength
sampled with 39 micro-windows (Kerzenmacher et al., 2008; Sheese et al., 2016). The main
interfering species in this region is $O_3$, with smaller contributions from $CO_2$, $H_2O$, and $COF_2$.
MIPAS operated onboard the Envisat satellite during 2005 – 2012 in a sun-synchronous orbit with
equator crossing at 10 am and 10 pm local time. MIPAS measured limb emission spectra covering
4.15 to 14.6 μm wavelength using a Fourier transform spectrometer. MIPAS primarily observed
altitudes from 6 to 68 km, with periodic (one day in ten) observations extending into the
thermosphere (~150 km). The MIPAS NO product is reported at 1 km intervals, but has a vertical
resolution of 5 - 15 km, except within the upper mesosphere outside polar winter where the
resolution degrades up to 20 km. NO emission measured at 5.3 μm was used to retrieve NO volume
mixing ratios (VMR) (Funke et al., 2005, Bermejo-Pantaléon et al., 2011). The mixing ratios were
converted to number densities (ND, molecules $cm^{-3}$) using temperatures derived from 15 μm
emissions below 100 km and from 5.3 μm above (jointly retrieved with NO). This work uses data
version V5r_NOwT_622. Bender et al (2015) report NO measurements comparisons including
ACE, MIPAS, the SCanning Imaging Absorption spectroMeter for Atmospheric CHartographY
(SCIAMACHY) instrument, and the sub-millimeter radiometer (SMR) satellite instrument. They
found mean differences of 30 to 100%, depending on latitude, season, and altitude. While this
work does not include SCIAMACHY or SMR results, the agreement of these observations with
SOFIE can be inferred through inspection of Bender et al (2015).
**2. SOFIE Observations**

SOFIE uses solar occultation to measure vertical profiles of temperature, five gaseous

species ($O_3$, $H_2O$, $CO_2$, $CH_4$, and NO), polar mesospheric clouds (PMC), and meteoric smoke
(Gordley et al., 2009; Hervig et al., 2009). Spacecraft sunset measurements always occurred in the
Southern Hemisphere (SH), with sunrise in the Northern Hemisphere (NH), for the measurements
during 2007-2017 used here. In late 2018 this changed with sunsets switching to the NH. NO
measurements are accomplished using broadband (~2% filter width) measurements centered at
5.32 μm wavelength. Gomez-Ramirez et al. (2013) provide a detailed description of the SOFIE
NO measurements, signal corrections, and retrievals. The photo conductive detector experiences
a response oscillation due to the thermal shock of transitioning the field-of-view (FOV) from dark
space to the sun, at the start of each observation. This thermal response artifact was successfully
corrected in ground processing, as discussed in detail by Gomez-Ramirez et al. (2013). The
subsequent NO retrievals are conducted in terms of VMR, for altitudes of ~30 to 149 km. The
SOFIE FOV subtends ~1.5 km vertically, but retrieved NO has a coarser effective vertical
resolution (~2.5 km) due to measurement noise and retrieval errors. Gomez-Ramirez et al.
compared SOFIE version 1.2 NO profiles to coincident ACE measurements for altitudes from 87
- 105 km, showing negligible differences for SH SOFIE measurements (spacecraft sunset) and
~18% differences in the NH (sunrise). SOFIE retrieves temperatures (T) from 17 - 100 km altitude,
and T from the mass spectrometer incoherent scatter (MSIS) model are used above 100 km (see
Marshall et al., 2011). Because VMR requires knowledge of air density (and thus T), the retrieved
VMR likely contain large errors above 100 km due to MSIS T uncertainties. SOFIE VMR are thus
converted to ND in post processing, using the appropriate T/P values (SOFIE or MSIS). NO ND
has the advantage of being independent of T, and thus is recommended for use above 100 km
(available online).

SOFIE NO profiles contain values that indicate missing data ($-10^{24}$), which imply that the

signal was either not measured or contained artifacts that rendered it unusable. There are also
values which indicate a good measurement, but an unsuccessful retrieval ($10^{-14}$ in VMR). These
instances correspond to cases where the simulated signal considering interfering gases was greater
than the observed signal. These situations clearly indicate errors in the interference, and/or the
measured signals. In V1.3, the unsuccessful retrievals were included in vertical smoothing of the
NO VMR profile prior to output, which resulted in large errors in the two points above and below
the unsuccessful layer. These values were filtered (set to the missing data value of $-10^{24}$) in post-
processing, along with points associated with PMCs, which have erroneously increased NO (see
details below). PMCs are clearly identified in SOFIE profiles using multi-wavelength observations
as described in Hervig et al. (2009). The filtered profiles were then smoothed by box-car averaging
on a 3 km vertical grid (see Figure 1a). The filtered and smoothed V1.3 NO profiles are available
(as a mission data file, SOFIE_L2m_2007135_2017026_NO_den_filt_sm_01.3.nc) on the SOFIE
webpage (sofie.gats-inc.com).

Figure 1b shows the fraction of successful SOFIE NO measurements as a function of

altitude for SOFIE spacecraft sunrise and sunset. Between ~45 and 80 km, sunrises are successful
less than 20% of the time, while sunsets are successful more than 50% of the time. This is
comparable to ACE, which has a similar fraction of retrieval success at these heights, although no
appreciable difference between spacecraft sunrise and sunset (Figure 1b). MIPAS has very few
unsuccessful NO retrievals (<3%), and only reports the valid results. The often low fraction of
good NO results below ~80 km should be born mind when using the SOFIE (and ACE) NO
products.

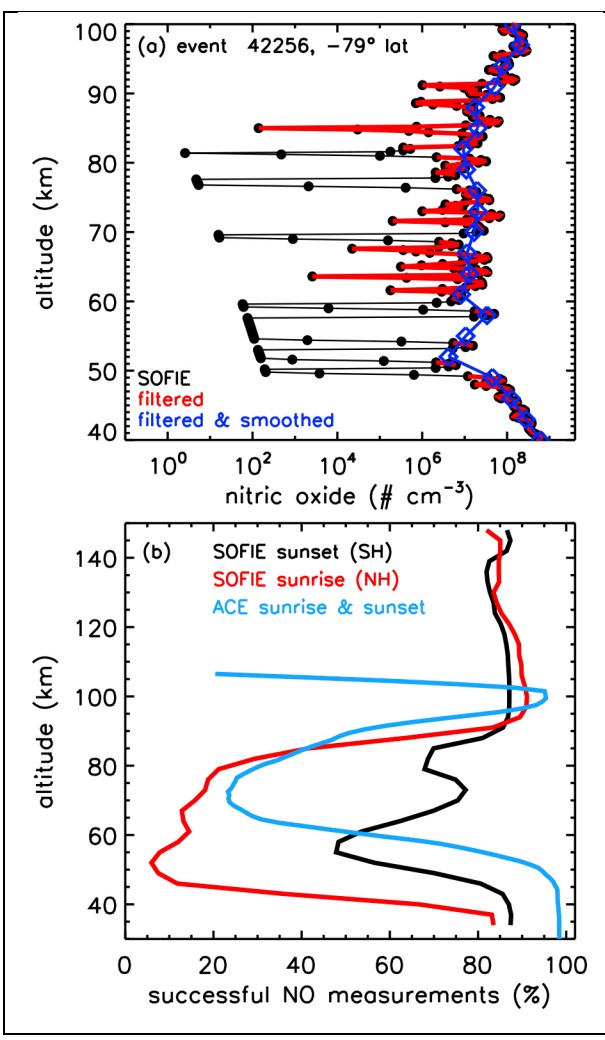

**Figure 1.** a) Example SOFIE NO retrieval from March 12, 2011, showing the original profile, the profile with erroneous values filtered (see text), and the filtered profile smoothed to 3 km spacing. b) The percentage of successful NO retrievals vs. altitude for SOFIE sunrise and sunset observations. ACE results are similar for sunrise and sunset, and are shown here for all measurements combined. Note that MIPAS only reports successful retrievals.

## 2.1. Uncertainty Analysis

The SOFIE NO uncertainty analysis presented here is an extension of the analysis described in Gomez-Ramirez et al. (2013). Retrieved NO error mechanisms can be categorized as due either to the SOFIE measurements, or to the signal simulations used in the retrievals. Simulation uncertainties include modeling errors, the representation of instrument characteristics (e.g., relative spectral response (RSR)), and the description of interfering gases and aerosols.

It is useful to first understand the relative signal contributions from interfering gases and aerosols in the SOFIE NO bandpass, as these can be the largest error sources. Figure 2 shows calculated signals considering polar summer conditions. The signal is due entirely to NO above

~85 km, with the main interference at lower altitudes coming from $H_2O$, $CO_2$, and $O_3$. $H_2O$
interference is removed using SOFIE $H_2O$ measurements which cover ~20 to 95 km altitude and
have uncertainties of ~15% (Rong et al., 2010). $CO_2$ is described using model results (Garcia et
al., 2007) which have uncertainties of <5%. $O_3$ interference is removed using SOFIE $O_3$ retrievals
that span ~55 - 110 km with uncertainties of <10% (Smith et al., 2013). Climatological $O_3$ is used
below 55 km, which can have large uncertainties. Fortunately the $O_3$ contribution to the SOFIE
NO signal is small at these heights (Figure 2). The upcoming SOFIE version (V1.4) will use new
SOFIE $O_3$ retrievals that extend down to ~15 km altitude. Interference from stratospheric sulfate
aerosols (SSA) is negligible above ~30 km, where NO is retrieved.

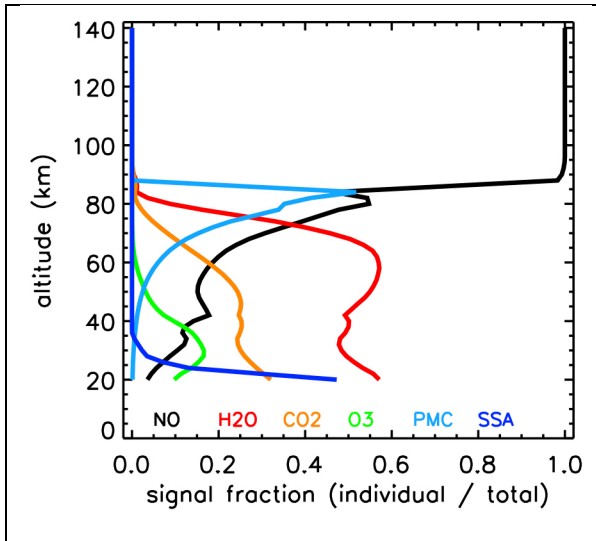

**Figure 2.** Relative contribution of various gases, PMCs (a layer from 81 - 87 km, centered at 84 km), and stratospheric sulfate aerosols (SSA), in the SOFIE 5.32 µm band used to measure NO. The results were simulated using average conditions near 66°S latitude in summer.


PMCs, which appear during polar summer, can contribute a large fraction of the total

SOFIE NO signal at PMC heights (~80 - 90 km). The example in Figure 2 is for a moderate PMC,
which contributes ~50% of the total signal near 84 km. This example also illustrates that the PMC
signal can extend from 20 to 30 km below the PMC layer, because the tangent path view includes
a contribution from altitudes above. PMC interference is not corrected during the retrievals in V1.3
(it will be in V1.4). As an interim step, the portion of NO profiles contaminated by PMCs (75 - 89
km when PMCs were present) was filtered (i.e., set to missing) in existing V1.3 profiles, for the
new V1.3 SOFIE data file described above. The artificial increase in retrieved NO when PMCs
are present is illustrated by comparing concurrent profiles with and without PMCs present, where
the contamination is obvious at ~80 to 90 km (Figures 3a and 3b). NO can be erroneously increased
by factors of 10 or more by PMC contamination (Figure 3c), and it is thus imperative to not use
NO when PMCs are present. Note that this effect is typically worse in the NH where PMCs
typically have greater volume density (e.g., Hervig et al., 2009). It is therefore recommended to
either use the new V.13 file, or ensure that PMC profiles are screened using the reported SOFIE
PMC observations (Hervig et al., 2009). Because PMC-induced errors occur only during polar
summer and not necessarily in every profile, PMC induced NO errors are not included in the total
uncertainty estimates below.

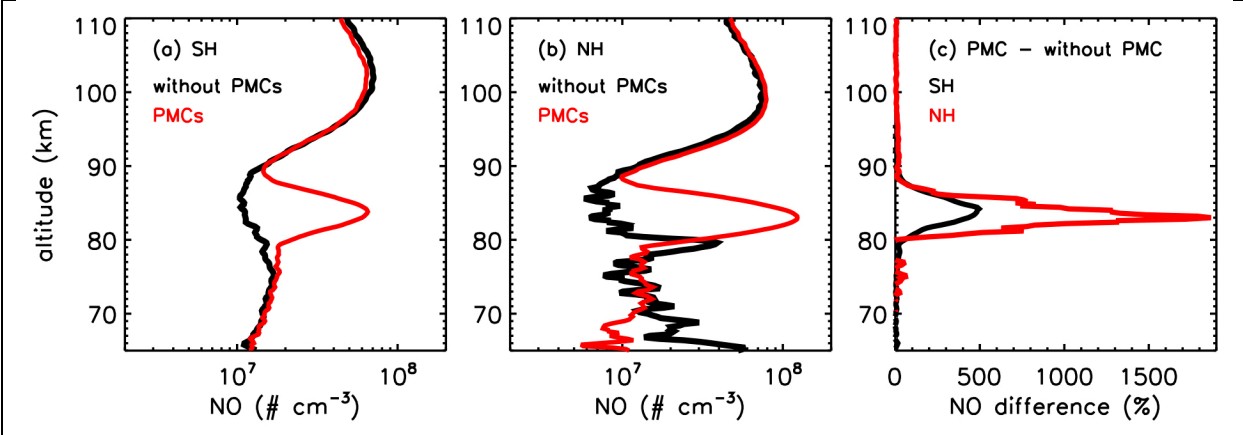

**Figure 3.** Comparison of average NO profiles during polar summer (-30 to 60 days from solstice, during 2007 to 2013) with and without PMCs present, for the a) SH and b) NH. c) Difference in average NO ND for the profiles with and without PMCs, for both hemispheres.


The main error sources in retrieved NO are summarized in Table 1 for a range of altitudes.

The largest measurement errors are due to noise and the thermal response correction, which is
larger for sunrise observations than in sunsets (see Gomez-Ramirez et al. (2013) for details). The
remaining errors are in the category of measurement interpretation as encompassed by model
simulations of the SOFIE signal. Errors in the interfering gases (measured or modeled) were taken
from the relevant publications, as discussed above. Each error mechanism was imposed in the V1.3
SOFIE retrieval algorithm to determine the uncertainty induced in retrieved NO ND. The V1.3
SOFIE forward model uses HITRAN 2004 line parameters, which are estimated to have ~7%
systematic uncertainties for NO near 5.32 μm. Altitude registration errors are estimated to be ~100
m (Marshall et al., 2011). While errors in temperature propagate directly into NO VMR, they do
not affect ND, which is a strong argument for using ND in the thermosphere where SOFIE does
not measure temperatures. The uncertainties in retrieved NO are summarized at key altitudes in
Table 1 for each mechanism, along with the total uncertainty. The largest four error sources are
shown versus height in Figure 4, where it is clear that water vapor interference errors dominate
below ~90 km, for both sunrise and sunset. For sunset measurements NO ND errors are dominated
by noise above ~100 km. Sunrise NO errors are dominated by the thermal response correction
above ~90 km, as discussed by Gomez-Ramirez et al. (2013).

**Table 1.** Uncertainty (%) in retrieved NO number density versus altitude due to various random (R) and systematic (S) error mechanisms. Two values are listed when they were different for sunrise / sunset.

| Error Source | Altitude (km) | | | | | |
|---|---|---|---|---|---|---|
| | 140 | 120 | 100 | 80 | 60 | 40 |
| Altitude Registration (S) | 1 | 2 | 5 | 10 | 5 | 2 |
| $H_2O$ Interference (S) | 0 | 0 | 1 | 30 | 30 | 10 |
| $CO_2$ Interference (S) | 0 | 0 | 1 | 3 | 5 | 3 |
| $O_3$ Interference (S) | 0 | 0 | 0 | 1 | 3 | 10 |
| Line Strengths (S) | 7 | 7 | 7 | 7 | 7 | 7 |
| Relative Spectral Response (S) | 5 | 5 | 5 | 5 | 5 | 5 |
| Field-of-View (S) | 2 | 3 | 4 | 4 | 3 | 3 |
| Forward Model (S) | 3 | 3 | 3 | 3 | 3 | 3 |
| Signal Noise (R) | 40 | 20 | 10 | 10 | 5 | 3 |
| Thermal Response Correction (R) | 30 / 15 | 30 / 15 | 30 / 10 | 20 / 5 | 10 / 3 | 5 / 3 |
| **Total** (root sum squared) | 51 / 44 | 37 / 27 | 34 / 18 | 40 / 35 | 34 / 33 | 18 / 18 |


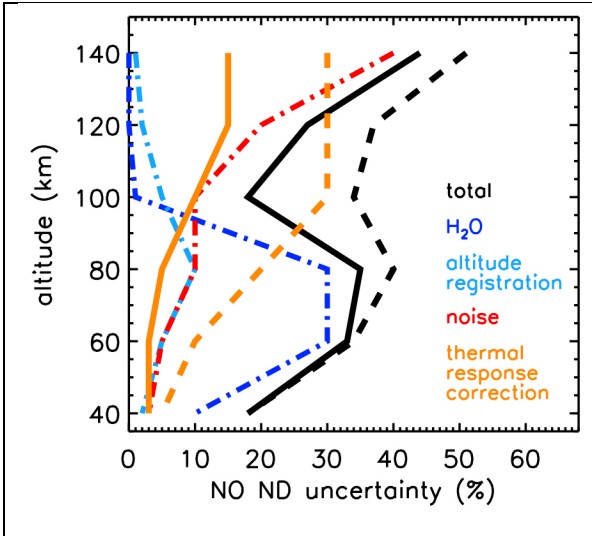

**Figure 4.** SOFIE NO uncertainties vs height. Results are shown for the four largest error mechanisms (by color), and for the total (random plus systematic) uncertainty. Values are as given in Table 1. Dashed curves represent sunrise and solid curves indicate sunset results. Dot-dash lines apply to both sunrise and sunset.

## 4. Measurement Comparisons

Time separation is important in the measurement comparisons because NO abundance can have a strong diurnal dependence, with more than 10% per hour changes in ND near local sunrise or sunset, depending on altitude, latitude, and season (e.g., Siskind et al., 2019). This effect can be managed in the comparisons by 1) keeping the measurement separations as small as possible, or 2) applying a modeled diurnal correction to measurements that are separated in time. Removing diurnal dependence using a model description was determined to induce unacceptably large uncertainties, in part because the model results are dependent on transport as well as photochemistry. The first approach was therefore adopted here, finding coincident measurement pairs for maximum separations of 2 hours UT, 4° latitude, and 20° longitude. Note that 20° longitude corresponds to ~1.3 hours in local time. These coincidence criteria insured that average measurement separations were less than one hour. Note that when this work mentions sunrise or sunset (for SOFIE and/or ACE) that it always refers to the view from orbit. SOFIE spacecraft sunset is always Earth sunrise (and vice versa), due to the retrograde polar orbit. ACE can have

varying correspondence between sunset or sunrise as viewed from orbit or Earth, and thus it is
important to track LT in the comparisons. Finally, the comparisons shown below include SOFIE
profiles with PMCs, and the results do not change when excluding profiles with PMCs. This is
because SOFIE NO results used here have been filtered at PMC heights when PMCs were present
(see Section 2), and because the MIPAS and ACE NO measurements are not affected by PMC
contamination (Funke et al., 2005; Kerzenmacher et al., 2008). SOFIE - ACE coincidences are
illustrated in Figure 5 including a summary of the coincidence statistics, and SOFIE - MIPAS
coincidences are shown in Figure 6.

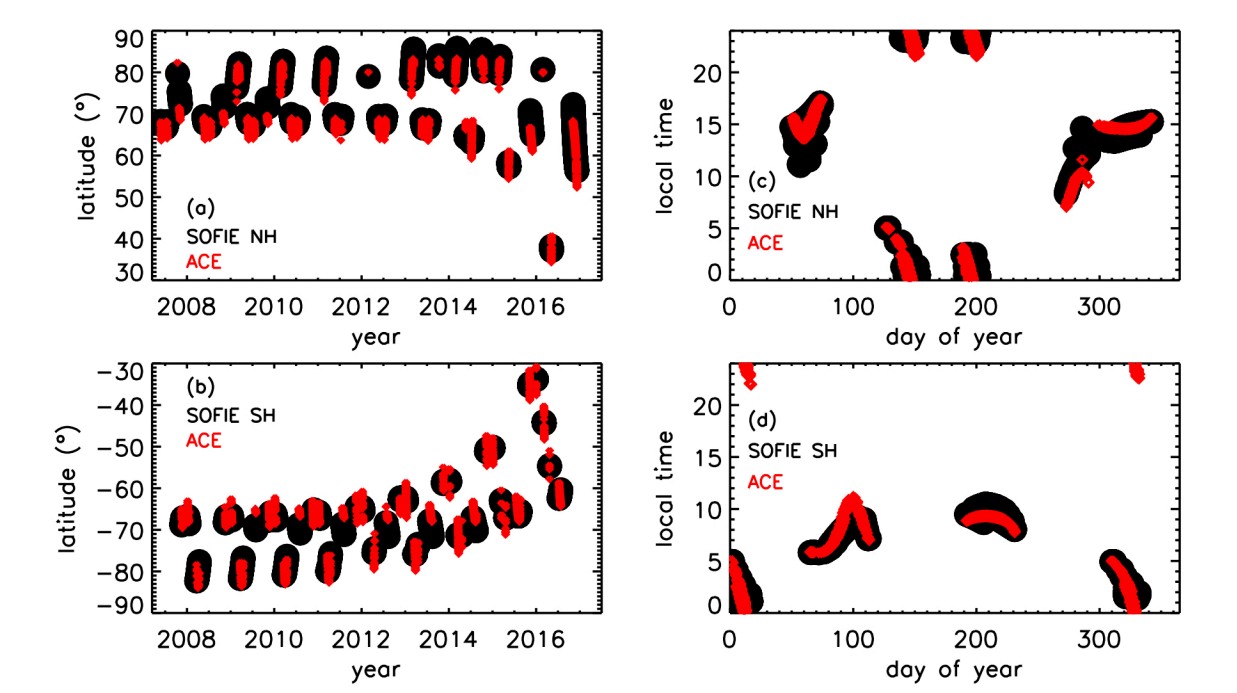

**Figure 5.** Summary of SOFIE - ACE coincidences. Measurement latitude vs. year in the a) NH (SOFIE sunrise; local sunset) and b) SH (SOFIE sunset; local sunrise). Measurement LT versus day of year in the c) NH and d) SH. There were 2968 coincidences in the NH with average separations of 0.7 hours, 1.7° latitude, and 8.0° longitude.  There were 2473 coincidences in the SH with average separations of 0.6 hours, 2.3° latitude, and 8.0° longitude.

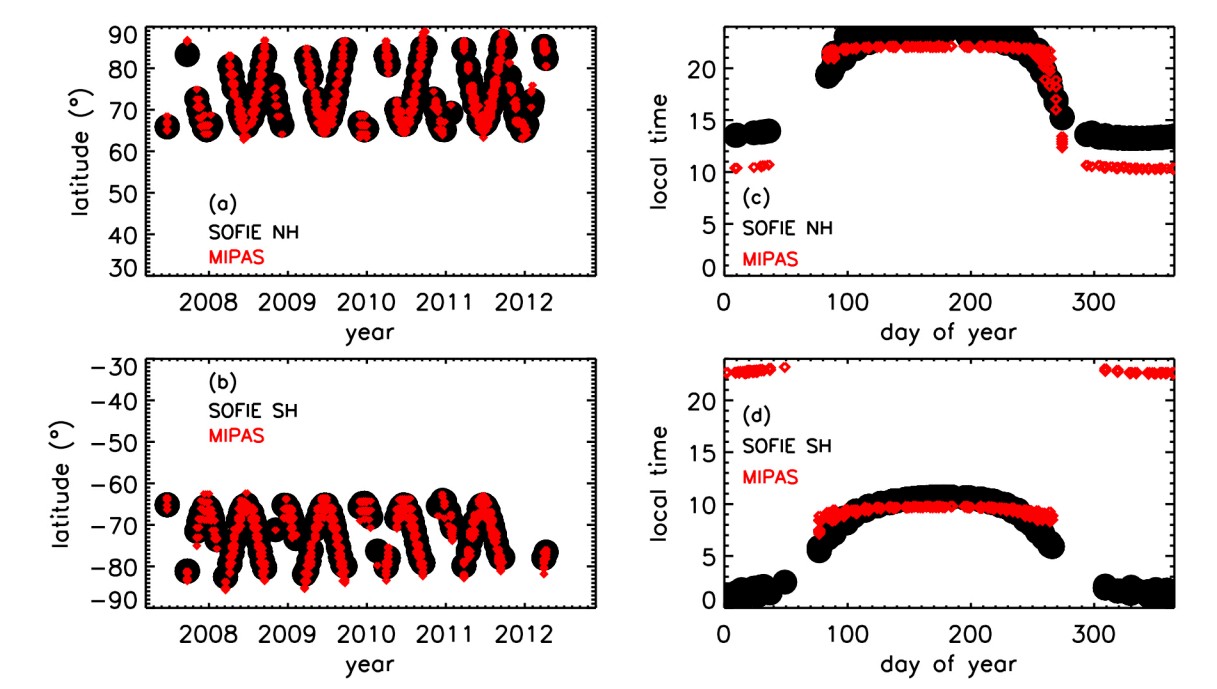

**Figure 6.** Summary of SOFIE - MIPAS coincidences. Measurement latitude vs. year in the a) NH (SOFIE sunrise; local sunset) and b) SH (SOFIE sunset; local sunrise). Measurement LT versus day of year in the c) NH and d) SH. The NH had 894 coincidences with average separations of 0.9 hours, 1.3° latitude, and 9.6° longitude. The SH had 985 coincidences with average separations of 0.8 hours, 1.4° latitude, and 8.7° longitude. Note that the MIPAS solar zenith angles ranged from 82 - 95° for the SH SOFIE comparisons and 84 - 94° for the NH comparisons, which is near local sunrise (or sunset).

SOFIE, ACE, and MIPAS have effective vertical resolution of roughly 2.5, 3.5, and >5km,
respectively, despite differences in the FOVs and reported vertical spacing. For the comparisons
shown here, the ACE and MIPAS results were interpolated to the SOFIE 3 km vertical scale, with
no additional smoothing applied. Note that the results below are essentially unchanged if the NO
profiles are interpolated to either the ACE or MIPAS vertical scales instead. Comparison of NO
vertical profiles are shown in Figure 7 for SOFIE vs. ACE, and in Figure 8 for SOFIE vs. MIPAS.
The comparisons are shown as average profiles, mean and root-mean-square (RMS; i.e. random
plus systematic) differences, and the number of points used in the comparison at each altitude.
SOFIE - ACE mean differences are within 50% for altitudes from ~50 to 107 km in both the SH
and NH (Figures 7b and 7d). SOFIE - MIPAS differences are within ~50% for ~55 - 140 km in
the SH (Figure 8). The NH MIPAS comparison indicates larger differences than in the SH, but
with some similarities in the dependence on height (e.g. SOFIE > MIPAS near 140 km). The
SOFIE - MIPAS comparison above ~130 km in the SH (~140 km in the NH)  indicates an
increasing bias with SOFIE suggesting higher NO. Siskind et al. (2019) noted a similar bias from
indirect comparisons of SOFIE with the Student Nitric Oxide Explorer (SNOE) results. Note that
the number of measurement pairs used in the comparisons is fairly consistent in height for the SH
(SOFIE sunset), in both the ACE and MIPAS comparisons (Figures 7c and 8c). The NH (SOFIE
sunrise) comparisons, however, have very few valid measurements between ~50 and 80 km
(Figures 7f and 8f), due to the lack of good SOFIE (and sometimes ACE) results at these altitude
for sunrise.

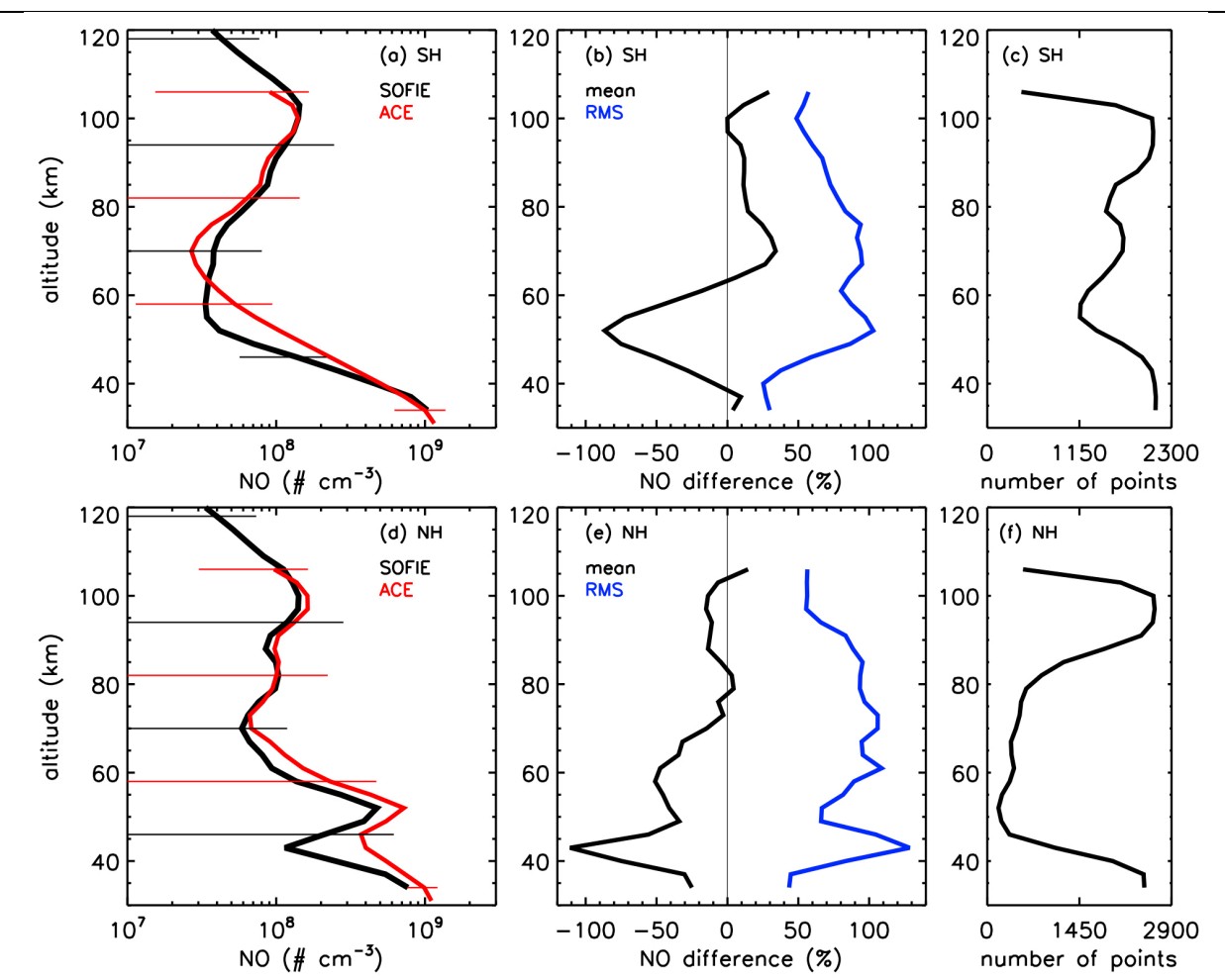

**Figure 7.** Comparison of SOFIE and ACE NO number density profiles, for the coincidences shown in Figure 5. Comparisons in the SH (SOFIE spacecraft sunset; local sunrise) as a) average profiles, b) mean and RMS differences, and c) number of points in the comparison at each altitude. Comparisons in the NH (SOFIE sunrise; local sunset) as d) average profiles, e) mean and RMS differences, and f) number of points in the comparison. Horizontal lines on the average NO profiles indicate standard deviations.


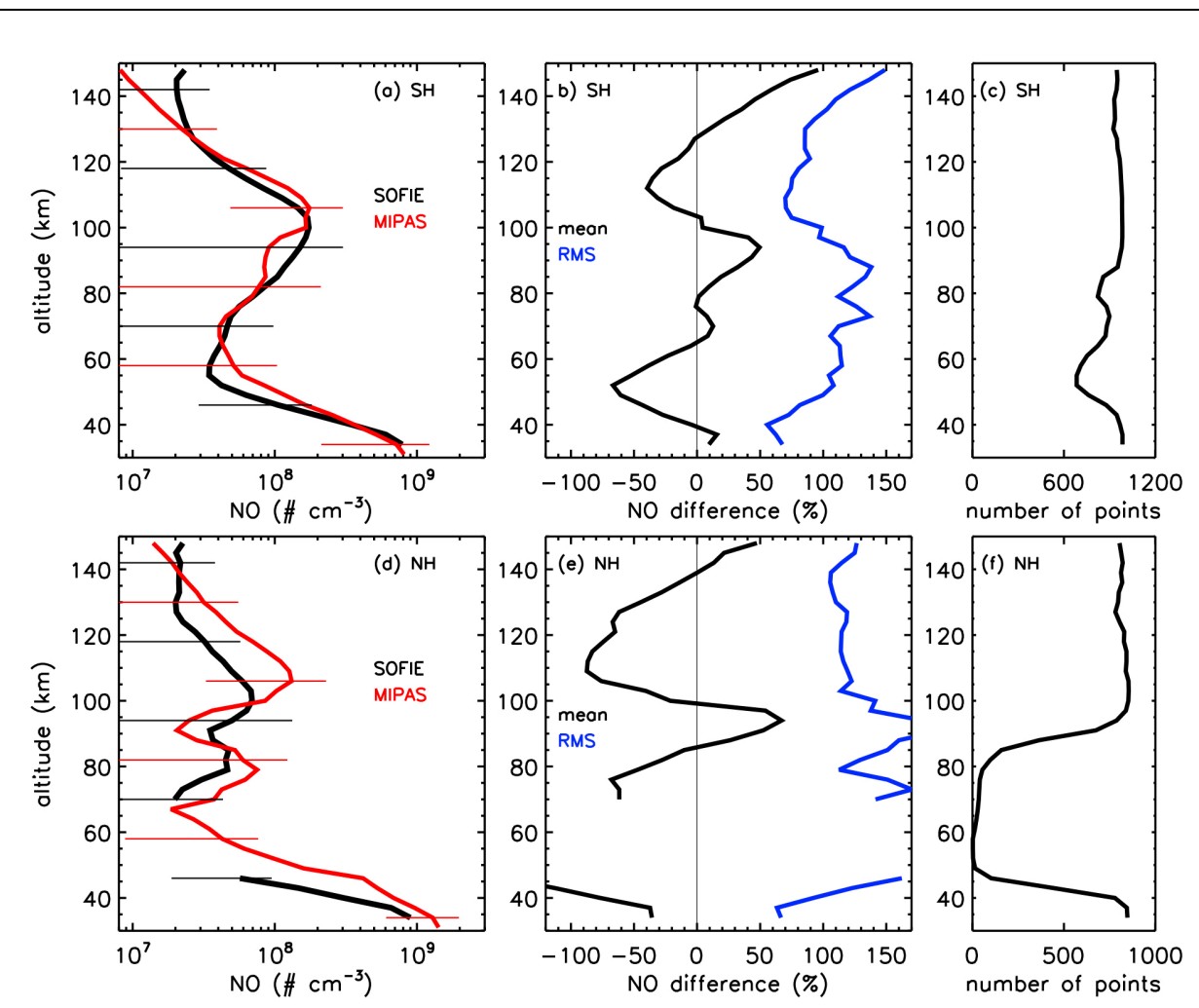

**Figure 8.** Comparison of SOFIE and MIPAS NO vertical profiles, for the coincidences shown in Figure 6. Comparisons in the SH (SOFIE spacecraft sunset; local sunrise) as a) average profiles, b) mean and RMS differences, and c) number of points in the comparison at each altitude. Comparisons in the NH (SOFIE sunrise) as d) average profiles, e) mean and RMS differences, and f) number of points in the comparison. Mean NO and NO differences are only shown when there were more than 30 points in the comparison. Horizontal lines on the average profiles indicate standard deviations.


Comparing the SOFIE - ACE and SOFIE - MIPAS mean differences shows notable

similarities in both the height dependence and magnitude of the differences, especially in the SH
(Figure 9a). In particular, SOFIE NO is consistently ~50% or more lower than ACE and MIPAS
near the stratopause (~50 km) in both the SH and NH (Figure 9). These similarities suggest the
presence of a systematic error in SOFIE, although a potential error mechanism has not yet been
identified. It should be noted that diurnal variations in NO, which are strongest in the stratosphere
and thermosphere, can determine that occultation measurements are viewing through strong spatial
gradients along the tangent path. The impact of such gradients has not yet been quantified, but
chould appear as a systematic bias in retrieved NO. The measurement coincidences were close
enough in LT that diurnal variations should be a small part of the comparison differences. It is
rather the increased SOFIE errors for sunrise (NH) that explain differences in the SOFIE - ACE
and SOFIE - MIPAS comparisons between the NH and SH. Note that the comparisons in the NH
additionally indicate that MIPAS NO is greater than ACE, particularly below ~90 km (Figure 9b),
a difference that was also reported by Bender et al. (2015).

Time series of monthly zonal mean NO at selected altitudes are compared for the SOFIE -

ACE coincidences in Figure 10, and for the SOFIE - MIPAS coincidences in Figure 11. These
time series indicate good agreement on the timing and magnitude of NO variations, despite
systematic differences at certain altitudes. To better quantify the agreement concerning time
variations, linear correlation coefficients were determined for each height in the SOFIE - ACE and
SOFIE - MIPAS comparisons. Results in the SH (Figure 12a) show a strong correlation between
SOFIE and ACE or MIPAS for altitudes below ~130 km. Results in the NH (Figure 12b) indicate
a significant correlation between SOFIE and ACE for 90 - 107 km. The NH SOFIE - MIPAS
comparisons also indicate a high correlation for ~90 - 110 km. Note that the correlations were not
determined in the NH for ~50 to 85 km because there were very few SOFIE NO retrievals (e.g.
Figures 10e and 11g).

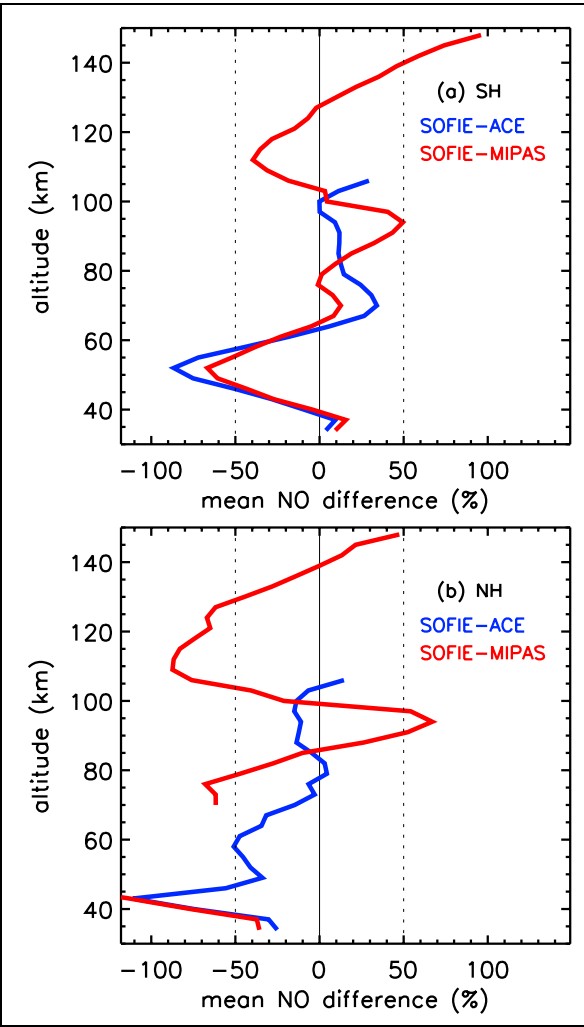

**Figure 9.** Mean NO differences versus height for comparisons of SOFIE with ACE and MIPAS in the a) SH (SOFIE sunset) and b) NH (SOFIE sunrise). The mean differences are as shown in Figures 6 and 7. Mean NO and NO differences are only shown when there were more than 30 points in the comparison.


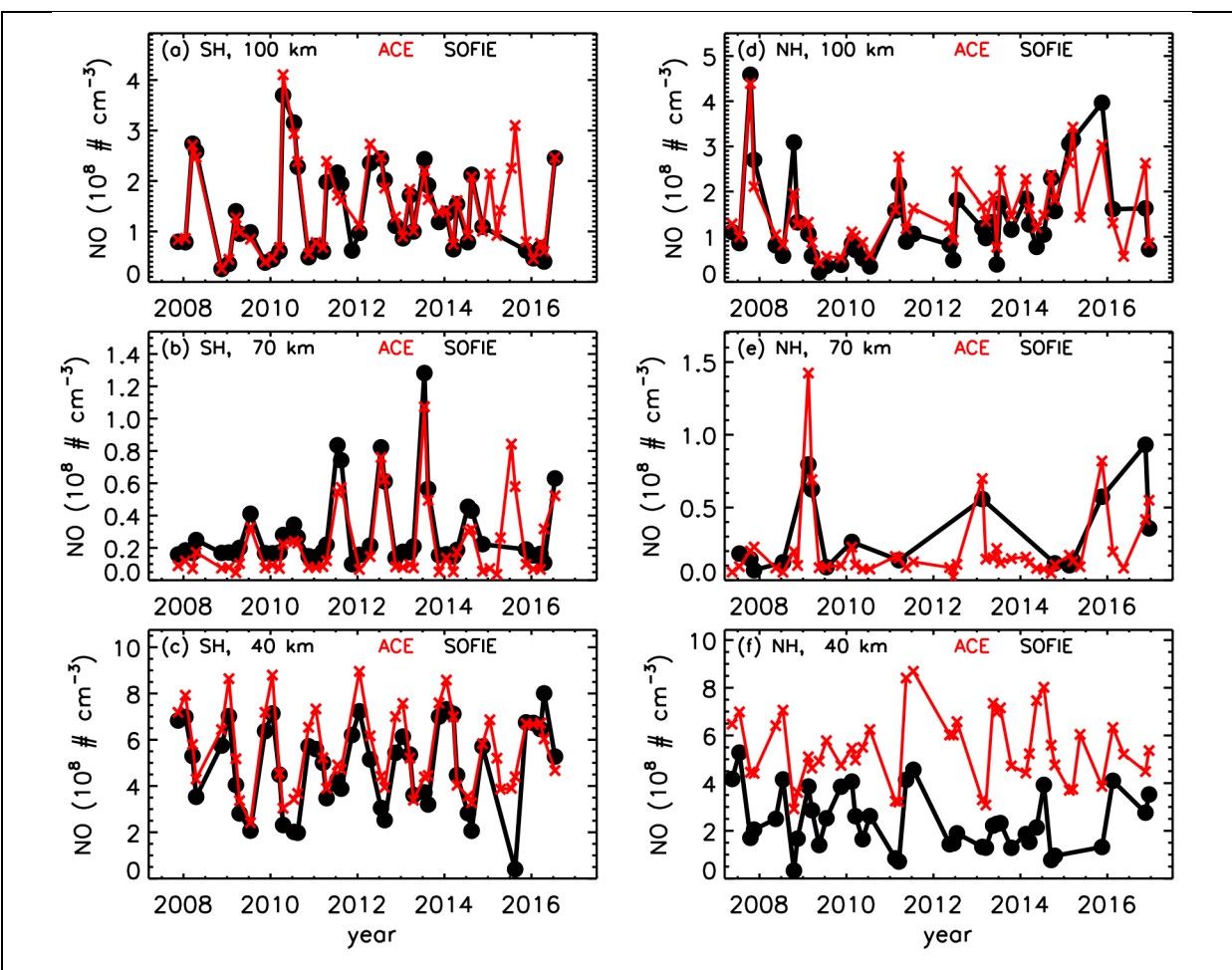

**Figure 10.** Comparison of SOFIE and ACE NO time series as monthly zonal means, for the coincidences shown in Figure 5. SH results are shown for a) 100 km, b) 70 km, and c) 40 km altitude. NH results are shown for d) 100 km, e) 70 km, and f) 40 km altitude.


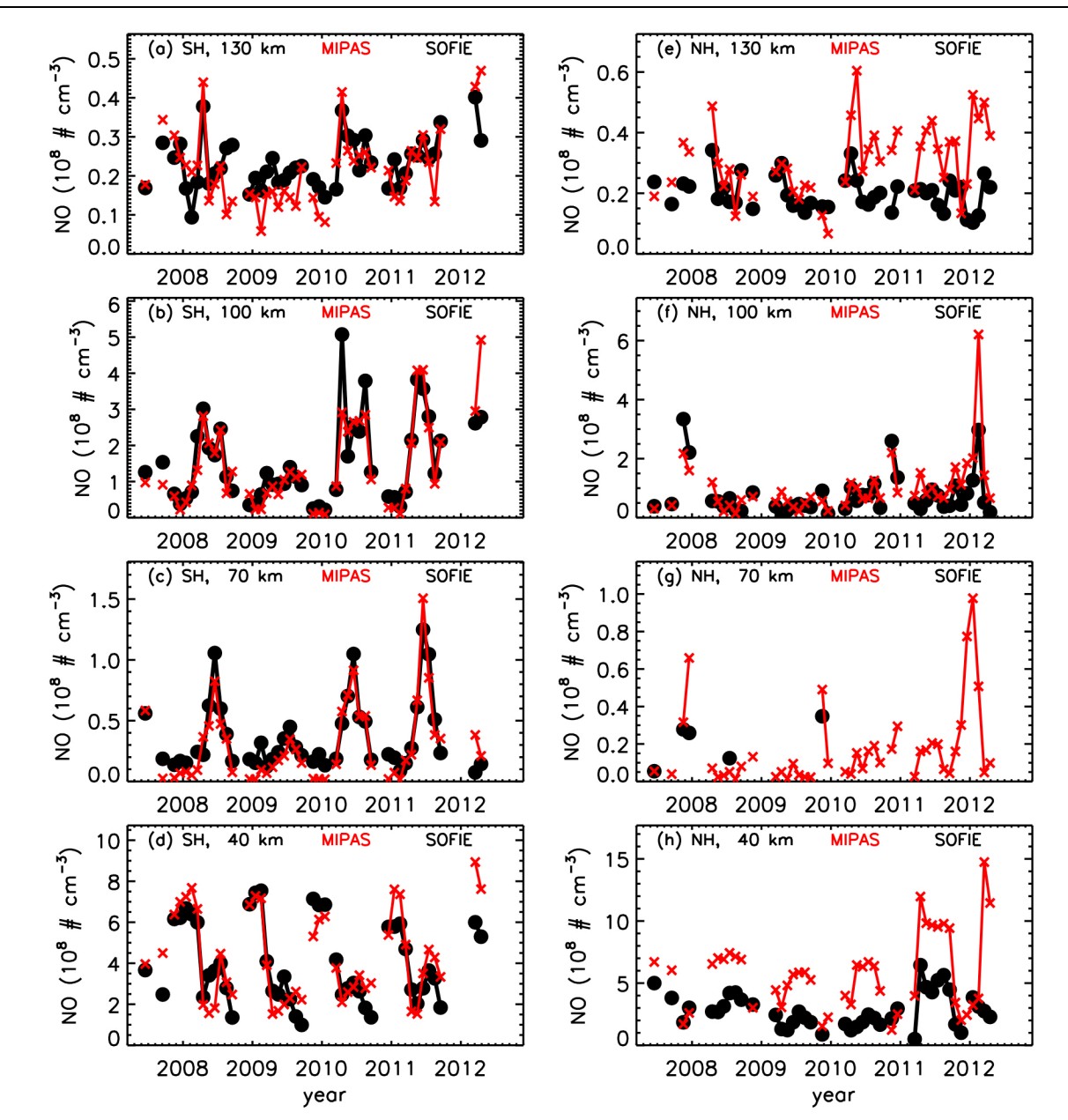

**Figure 11.** Comparison of SOFIE and MIPAS NO time series as monthly zonal means, for the coincidences shown in Figure 6. SH results are shown for a) 130 km, b) 100 km, c) 70 km, and d) 40 km altitude. NH results are shown for e) 130 km, f) 100 km, g) 70 km, and h) 40 km altitude.

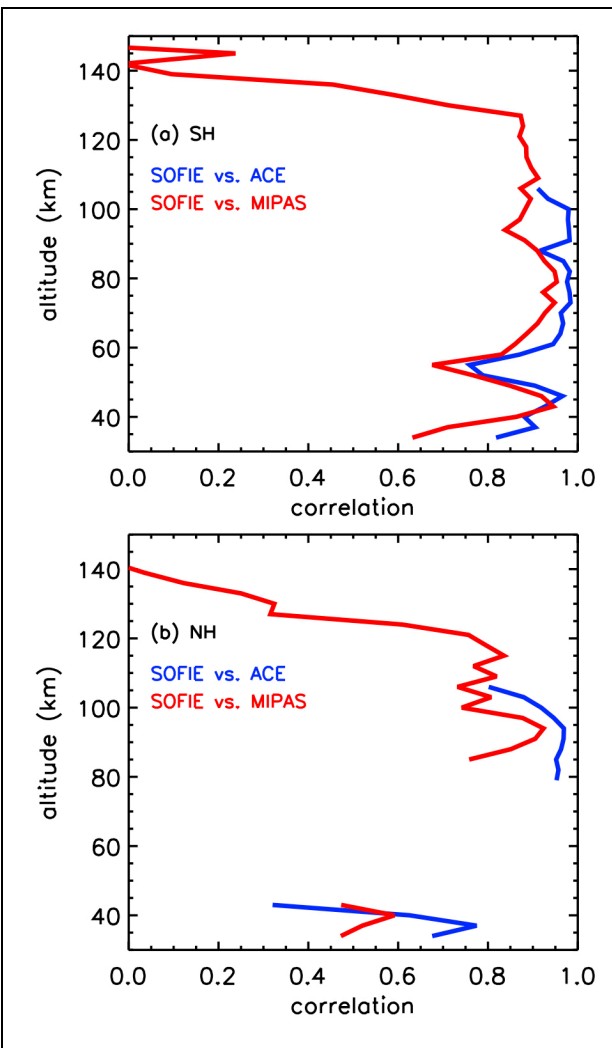

**Figure 12.** SOFIE - ACE and SOFIE - MIPAS correlation coefficients for comparison of monthly mean NO time series (as in Figures 10 and 11). Results are shown versus height in the a) SH and b) NH. Note that results are only shown when more than half of the monthly mean points were valid for both instruments, which was primarily a concern for the NH below ~80 km. Where results are shown, there were typically more than 40 points in the comparison, for which the 95% significance level is a correlation coefficient of ~0.3 or greater.

## 5. Summary


Comparisons of SOFIE NO with coincident measurements from ACE and MIPAS indicate
mean differences of less than ~50% for altitudes from roughly 50 to 105 km for SOFIE spacecraft
sunrise, and ~50 to 140 km for SOFIE sunsets. Comparisons of NO time series show significant
correlation between SOFIE and either ACE or MIPAS for altitudes of ~40 - 130 km in the SH,
indicating that measured NO variability is robust. Correlations were significant in the NH for ~90
to 130 km, but not at lower heights due to the sparse SOFIE results in that altitude range. SOFIE
uncertainties increase below ~85 km due primarily to interfering $H_2O$ absorption and signal
correction errors. These effects are sufficiently large in SOFIE sunrise measurements that retrieved
NO is only reliable below ~80 km during enhancement events (in <20% of the data), such as
downward transport due to a sudden stratospheric warming (e.g., Bailey et al., 2014). SOFIE
sunset signals have lower signal correction errors, and the retrieved NO is reliable in more than
half of the measurements below 80 km. SOFIE NO should not be used when PMCs are present
due to the often extreme contamination, and these instances were filtered (i.e. flagged as missing)
in the latest SOFIE V1.3 NO product which is available online.
**Acknowledgements.** This work was funded by the AIM mission through NASA Small Explorer
contract NAS5-03132, and through NASA Heliophysics Guest Investigator project
80NSSC19K0281 "Quantifying Drivers of Nitric Oxide Variability in the MLT Using SOFIE,
WACCM, and DART". SOFIE data are available online at sofie.gats-inc.com. ACE is funded by
the Canadian Space Agency with P. Bernath (University of Waterloo and Old Dominion
University) as the Mission Scientist. ACE data are available online at databace.scisat.ca. MIPAS
data are available online at share.lsdf.kit.edu/imk/asf/sat/mipas-export/.

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
