# Peer review of "Validation of SOFIE nitric oxide measurements"

_Atmospheric Measurement Techniques, 2019_

## Referee Comment (RC1) · Martin Friedrich (Referee) · 29 Mar 2019

The manuscript "Validation of SOFIE Nitric Oxide Measurements" by Hervig et al. represents a laudable effort to critically assess the quality of the NO data of the SOFIE experiment. In the very detailed analyses the contribution by water vapor, important below 85 km, emerges as an identified weakness of the data obtained by the SOFIE retrieval. Another significant discrepancy is between SOFIE and MIPAS above 120 km (Fig. 12). The fact that [NO] at sunrise and sunset in the mesosphere differ, is well established; why the differences between SOFIE vs. ACE (sunrise) and SOFIE vs. MIPAS (sunset) should also be different may have escaped me (Figs. 7 & 8). Similarly, I miss (or overlooked) a statement/suggestion why [NO] is apparently systematically different in the two hemispheres. Finally, I recommend to propose (or compose) a preliminary empirical model of NO considering the valuable findings that result from the present paper. Given that the above comments are addressed, I definitively recom-

mend publication.

---

## Referee Comment (RC2) · Anonymous Referee #2 · 8 Apr 2019

The paper describes the validation of the NO density retrieved from SOFIE against that retrieved from the MIPAS and ACE instruments. Since the SOFIE NO data has been used in a number of studies on the effect of particle precipitation on the atmosphere, this validation is both timely and important to the community. It should be published after minor revisions.

There are some general comments that the authors should address, as well as some minor corrections.

General comments:

1) The SOFIE NO density is validated against that measured by the MIPAS and ACE instruments. It is mentioned that the NO retrieved from the SCIAMACHY instrument were validated against MIPAS and the Odin Submillimetre Radiometer (SMR). However, it is not clear why these two data sets, SCIAMACHY amd SMR, were not used in

this validation. The authors should mention whey these data sets were excluded from the SOFIE validation.

2) It is mentioned in the text that the SOFIE NO density, and not the volume mixing ratio (vmr) should be used due to the use of MSIS temperatures above 100 km to convert to vmr. The reference given for the retrievals, Gordley et al, 2009, is focused on the PMC extinction, but does refer to the SOFIE Algorithm Theoretical Basis Document. It that document it is stated that: "Simulated signals are compared to the measurement, and the target gas mixing ratio, Q, is adjusted based on the derivative d-tau/dQ, which considers the previous attempt to match the measurement." This would indicate that it is the vmr that is the primary quantity being derived form the SOFIE measurement, and that is being converted to density using the measured/modelled temperatures. This should be clarified. One notes that these documents pre-date the NO retrievals, and reference should be made to any updated documentation of the NO retrieval process.

3) Related to comment 2, the MIPAS data use a logarithmic retrieval of vmr that will exclude negative values. This causes a net positive bias, particularly where the retrieved vmr values are low. Does SOFIE use a similar retrieval mechanism, and if not, would this explain some of the bias between the observations?

Minor corrections: Line 221. The word "determine" should be "determined".

Lines 231-237. It is stated that due to interfering absorption or signal corrections, some of the SOFIE NO data is not reliable. Are these unreliable data flagged in the data base? If not, then the authors should highlight this section as a major caution to users.

Lines 237-239. It says that instances where extreme contamination due to the presence of PMC are filtered in the latest SOFIE V1.3 NO online product. However, on line 124 it is stated that these instances are not filtered in V1.3 but will be in V1.4. This discrepancy should be rectified.

---

## Author Comment (AC1) · 2 May 2019

The comment was uploaded in the form of a supplement:
https://www.atmos-meas-tech-discuss.net/amt-2019-56/amt-2019-56-AC1-supplement.pdf

---

## Author Response (AR2)

**Author Response to Reviewer #1**

*Author responses are in italics.*

The manuscript "Validation of SOFIE Nitric Oxide Measurements" by Hervig et al. represents a laudable effort to critically assess the quality of the NO data of the SOFIE experiment. In the very detailed analyses the contribution by water vapor, important below 85 km, emerges as an identified weakness of the data obtained by the SOFIE retrieval. Another significant discrepancy is between SOFIE and MIPAS above 120 km (Fig. 12). The fact that [NO] at sunrise and sunset in the mesosphere differ, is well established; why the differences between SOFIE vs. ACE (sunrise) and SOFIE vs. MIPAS (sunset) should also be different may have escaped me (Figs. 7 & 8). Similarly, I miss (or overlooked) a statement/suggestion why [NO] is apparently systematically different in the two hemispheres. Finally, I recommend to propose (or compose) a preliminary empirical model of NO considering the valuable findings that result from the present paper. Given that the above comments are addressed, I definitely recommend publication.

*SOFIE spacecraft sunrise (sunset) always occurred in the North (South), for the 2007-2016 data used in this paper (in late 2018, this reversed due to orbit changes). This is the main reason that SOFIE NO measurements are different between hemispheres, and the explanation is two-fold. First is the natural diurnal variation in NO (as you mention), and second is that measurement errors are different for sunrise vs. sunset (as discussed in Section 2.1). We feel that the coincident measurements were close enough in LT that diurnal variations should be a small part of the differences. It is rather the increased SOFIE errors for sunrise (NH) that explain differences in the SOFIE - ACE and SOFIE - MIPAS comparisons in the NH and SH. We have added statements that clarify these points (start of Section 2; discussion of Fig. 9).*

*We would support an empirical NO model that includes SOFIE observations, and welcome any collaboration in this future endeavor, however we feel that this is beyond the scope of the present paper. We note that there are already several empirical models for extant NO datasets from SNOE (Marsh et al.), ODIN-SMR (Kivranta et al.) and SCIAMACHY (Bender et al.).*

**Author Response to Reviewer # 2**

*Author responses are in italics.*

The paper describes the validation of the NO density retrieved from SOFIE against that retrieved from the MIPAS and ACE instruments. Since the SOFIE NO data has been used in a number of studies on the effect of particle precipitation on the atmosphere, this validation is both timely and important to the community. It should be published after minor revisions.

There are some general comments that the authors should address, as well as some minor corrections.

General comments:

1) The SOFIE NO density is validated against that measured by the MIPAS and ACE instruments. It is mentioned that the NO retrieved from the SCIAMACHY instrument were validated against MIPAS and the Odin Submillimeter Radiometer (SMR). However, it is not clear why these two data sets, SCIAMACHY and SMR, were not used in this validation. The authors should mention whey these data sets were excluded from the SOFIE validation.

*We should have used these other data sets, and to be honest, there is not a good reason for this omission. At this point, however, adding the new data sets would substantially change (and delay) the paper. We feel that by relating the Bender et al (2015) paper to the current results, that one can get an idea of how SCIAMACHY and SMR agree with SOFIE. A comment to this effect was added.*

2) It is mentioned in the text that the SOFIE NO density, and not the volume mixing ratio (vmr) should be used due to the use of MSIS temperatures above 100 km to convert to vmr. The reference given for the retrievals, Gordley et al, 2009, is focused on the PMC extinction, but does refer to the SOFIE Algorithm Theoretical Basis Document. It that document it is stated that: "Simulated signals are compared to the measurement, and the target gas mixing ratio, Q, is adjusted based on the derivative d-tau/dQ, which considers the previous attempt to match the measurement." This would indicate that it is the vmr that is the primary quantity being derived from the SOFIE measurement, and that is being converted to density using the measured/modelled temperatures. This should be clarified. One notes that these documents pre-date the NO retrievals, and reference should be made to any updated documentation of the NO retrieval process.

*The main SOFIE NO reference is Gomez-Ramirez et al. (2013), and we have clarified the statement to this effect at the beginning of Section 2. The Reviewer is correct that the retrievals are conducted in terms of VMR, and that ND is determined in post-processing. This point is now clear in Section 2.*

3) Related to comment 2, the MIPAS data use a logarithmic retrieval of vmr that will exclude negative values. This causes a net positive bias, particularly where the retrieved vmr values are low. Does SOFIE use a similar retrieval mechanism, and if not, would this explain some of the bias between the observations?

*The SOFIE NO retrieval is conducted on linear VMR (see above), and does not allow negative values. For species with large dynamic range like NO, it is always an issue that systematic errors which impact the lowest VMR values will tend to induce a high bias. Still, if MIPAS has the same effect as SOFIE, then that may not be the explanation. We are hesitant to make a statement on this because the true extent of this effect is not completely understood. We are thinking about ways to mitigate this in the upcoming SOFIE data version (V1.4).*

Minor corrections: Line 221. The word "determine" should be "determined".

*This was changed.*

Lines 231-237. It is stated that due to interfering absorption or signal corrections, some of the SOFIE NO data is not reliable. Are these unreliable data flagged in the data base? If not, then the authors should highlight this section as a major caution to users.

*Comments were added in section 2 that addresses this point.*

Lines 237-239. It says that instances where extreme contamination due to the presence of PMC are filtered in the latest SOFIE V1.3 NO online product. However, on line 124 it is stated that these instances are not filtered in V1.3 but will be in V1.4. This discrepancy should be rectified.

*There are two approaches to dealing with the PMC contamination. In the new V1.3 file, data that had PMC contamination were filtered, by replacing the values with the missing data flag. In V1.4 we will implement an actual removal of PMC contamination in the retrievals. The text was changed to make this clear. Most of the new text related to this comment appears in sections 2 and 2.1.*

[revised manuscript text omitted]